# Viral Coinfection of Children Hospitalized with Severe Acute Respiratory Infections during COVID-19 Pandemic

**DOI:** 10.3390/biomedicines11051402

**Published:** 2023-05-09

**Authors:** Célia Regina Malveste Ito, André Luís Elias Moreira, Paulo Alex Neves da Silva, Mônica de Oliveira Santos, Adailton Pereira dos Santos, Geovana Sôffa Rézio, Pollyanna Neta de Brito, Alana Parreira Costa Rezende, Jakeline Godinho Fonseca, Fernanda Aparecida de Oliveira Peixoto, Isabela Jubé Wastowski, Viviane Monteiro Goes, Mariely Cordeiro Estrela, Priscila Zanette de Souza, Lilian Carla Carneiro, Melissa Ameloti Gomes Avelino

**Affiliations:** 1Microorganism Biotechnology Laboratory, Institute of Tropical Pathology and Public Health, Federal University of Goiás, 235 St. Leste Universitário, Goiânia 74605-050, GO, Brazil; 2State Emergency Hospital of the Northwest Region of Goiânia Governador Otávio Lage de Siqueira (HUGOL), Anhanguera Avenue, 14.527–Santos Dumont, Goiânia 74463-350, GO, Brazil; 3Neonatal ICU of Clinical Hospital of Federal University of Goiás/EBSERH, 1st Avenue Leste Universitário, Goiânia 74605-020, GO, Brazil; 4Molecular Immunology Laboratory, Goiás State University, Laranjeiras Unity Prof. Alfredo de Castro St., 9175, Parque das Laranjeiras, Goiânia 74855-130, GO, Brazil; 5Institute of Molecular Biology of Paraná (IBMP), Professor Algacyr Munhoz Mader St, 3775–Industrial City of Curitiba, Curitiba 81350-010, PR, Brazil; 6Departament of Pediatrics, Federal University of Goiás, Universitaria Avenue, Leste Universitário, Goiânia 74605-050, GO, Brazil

**Keywords:** viruses, SARI, comorbidities, epidemiology, sickness

## Abstract

The main pathogens of severe respiratory infection in children are respiratory viruses, and the current molecular technology allows for a rapid and simultaneous detection of a wide spectrum of these viral pathogens, facilitating the diagnosis and evaluation of viral coinfection. Methods: This study was conducted between March 2020 and December 2021. All children admitted to the ICU with a diagnosis of SARI and who were tested by polymerase chain reaction on nasopharyngeal swabs for SARS-CoV-2 and other common respiratory viral pathogens were included in the study. Results: The result of the viral panel identified 446 children, with one infected with a single virus and 160 co-infected with two or more viruses. This study employed descriptive analyses, where a total of twenty-two coinfections among SARI-causing viruses were identified. Thus, the five most frequent coinfections that were selected for the study are: hRV/SARS-CoV-2 (17.91%), hRV/RSV (14.18%), RSV/SARS-CoV-2 (12.69%), hRV/BoV (10.45%), and hRV/AdV (8.21%). The most significant age group was 38.1%, representing patients aged between 24 and 59 months (61 individuals). Patients older than 59 months represented a total of 27.5%, comprising forty-four patients. The use of oxygen therapy was statistically significant in coinfections with Bocavirus, other CoVs, Metapneumovirus, and RSV. Coinfections with SARS-CoV-2 and the other different coinfections presented a similar time of use of oxygen therapy with a value of (*p* > 0.05). In the year 2020, hRV/BoV was more frequent in relation to other types of coinfections, representing a total of 35.1%. The year 2021 presented a divergent profile, with hRV/SARS-CoV-2 coinfection being the most frequent (30.8%), followed by hRV/RSV (28.2%). Additionally, 25.6% and 15.4% represented coinfections between RSV/SARS-CoV-2 and hRV/AdV, respectively. We saw that two of the patients coinfected with hRV/SARS-CoV-2 died, representing 9.52% of all deaths in the study. In addition, both hRV/hBoV and hRV/RSV had death records for each case, representing 8.33% and 6.67% of all deaths, respectively. Conclusion: Coinfections with respiratory viruses, such as RSV and hBoV, can increase the severity of the disease in children with SARI who are admitted to the ICU, and children infected with SARS-CoV-2 have their clinical condition worsened when they have comorbidities.

## 1. Introduction

The main pathogens of severe respiratory infection in children are respiratory viruses, and the use of real-time polymerase chain reaction (PCR) technology allows a rapid and simultaneous detection of a wide spectrum of these viral pathogens, facilitating the diagnosis and evaluation of viral coinfection in severe respiratory syndromes [1].

Viral coinfections are present in severe acute respiratory infections (SARI) in a substantial proportion of children, with a rate of 14% to 44% [2,3]. During the pandemic period, the coinfections had a potential role in increased morbidity and mortality rates in the world [4].

Many respiratory viruses can cause coinfection, such as Human Metapneumovirus (hMPV), Respiratory Syncytial Virus (RSV), and Human Rhinovirus (hRV), and data on these viruses in the COVID-19 pandemic help in the diagnosis of patients, as well as determining which follow-up for the propaedeutic clinical signs presented may be of paramount importance in choosing the optimal antiviral therapy [4,5].

In the beginning of the COVID-19 pandemic, in March 2020, when the first cases of SARS-CoV-2 appeared in Brazil, the number of SARI registered was 68,100, and in 2021, there were 62,772 cases registered in children and adolescents [6]. 

In the city of Goiânia, capital of Goiás State, located in the center of western Brazil, 11,509 cases of SARI were recorded in 2020, 7911 by SARS-CoV-19, 22 by Influenza, 112 by other viruses, and 3419 by unspecified SARI, in the age groups <02 to ≤60. In 2021, 17,576 cases of SARS were reported, by SARS-CoV-2 (14,775), Influenza (57), other respiratory viruses (187), and unspecified (2530), in the age groups from <02 to ≤60. In the age group <2 to 19 years of age, the cases of SARI reported in the State of Goiás in the years 2020 and 2021 were 5078 cases in total, 1042 by SARS-CoV-2, 63 cases by influenza, 1002 by other respiratory viruses, and 2903 by unspecified SARI [7].

Blocking and closing borders between countries are strict health measures that help contain the spread of SARS-CoV-2 and decrease other respiratory infections caused by other viruses in children [8]. During the COVID-19 pandemic period, in the United Kingdom, a rate of 2% of viral coinfections associated with a wide variety of respiratory viruses was found [9].

This work was carried out with children and adolescents hospitalized in an intensive care unit with severe respiratory infection or with suspected COVID-19, with the objective of describing the viral coinfections detected, comparing the clinical epidemiology of children due to viral coinfection, and to determine whether viral coinfections contributed to disease severity.

## 2. Methods

### 2.1. Ethical Aspects and Hospitals Participating in the Study

All procedures and protocols for sample collection and processing were submitted and approved, under registration number 33540320.7.0000.5078 by the Research Ethics Committee of Hospital das Clínicas–GO of Federal University of Goiás, located in Goiânia-Goiás, Brazil. All parents of sick patients and voluntary donors signed the informed consent form.

The hospital unit responsible for the collection of samples and the availability of medical records for the study was the State Emergency Hospital of the Northwest Region of Goiânia Governador Otávio Lage de Siqueira (HUGOL).

### 2.2. Target Population

This study included 606 pediatrics patients with SARI patients due to viral contamination. The samples were collected in one hospital in a capital in the Center-West region of Brazil in the period from March 2020 to December 2021. We considered the following eligibility criteria: children (≤14 years old) admitted to emergency pediatric intensive care units (ICU). The coinfections and clinical aspects related to them were also analyzed such as: sex, age, comorbidities, fever, dyspnea, respiratory distress, wheezing, cough, chest X-ray, chest computed tomography, use of corticosteroids, use of antibiotics, salbutamol, antivirals, oxygen therapy, non-invasive ventilation (NIV), invasive mechanical ventilation (IMV), and the outcomes, for the five main coinfections mentioned. In addition, all subjects were tested by RT-qPCR (TaqMan—Thermo Fisher^®^, Boston, MA, USA, EUA).

### 2.3. Collection and Processing of Samples

For viral identification, 606 samples were collected using Rayon swabs of nasal secretion of the 606 hospitalized children. The swabs were stored in 15 mL Falcon tubes containing 3 mL of viral transport medium (VTM) and sent to the laboratory. The samples were transferred to tubes to transport samples of 5 mL containing 750 μL of TRIzol-LS reagent (Invitrogen^®^, New York, NY, USA, EUA) and 250 μL of the sample collected. Then, the tubes with the samples were stored in a freezer at −80 °C until the moment of ribonucleic acid extraction.

The amplification of the genetic material was performed with the kits of the Thermo Fisher Scientific (Applied Biosystems TrueMarkTM Respiratory I Combo Kit [A56284C] and Applied Biosystems TrueMarkTM Respiratory II Combo Kit [A56286C]). Both kits contained primer/probe sets specific to genomic regions of adenovirus, enterovirus, influenza A (Pan), influenza B, respiratory syncytial virus A/B, rhinovirus, human metapneumovirus, SARS-CoV-2 and primer/probe specific to conjoint coronavirus 229E, HKU1 coronavirus, NL63 coronavirus, OC43 coronavirus, parainfluenza virus 1, parainfluenza virus 2, parainfluenza virus 3, parainfluenza virus 4, and RNase P genomic regions.

The steps of experiments followed the manufacturer’s instructions. A cycle threshold between 8 and 35 was considered a detected virus (positive); for greater than 40, undetected viruses (negative); and between 35 and 40 required confirmations.

### 2.4. Statistical Analysis

All analyzed data were deposited in Excel, version 2016 (Microsoft Windows™, Washington, DC, USA) and the data analyses were performed in the R software (R Core Team, Vienna, Austria), version 4.2.1 with the dplyr library. While the statistical analyses were performed with the aid of the R software, descriptive analyses of absolute and relative frequencies were performed with the standard functions of R Studio. To verify the association of variables, Pearson’s chi-square test was used, and the Student’s test was used to compare the means. Statistical modeling with binomial logistic regression, odds ratio calculations, and confidence intervals (95%) were performed using the “mfx” libraries. All graphical analyses were built using the ggplot2 and plotly libraries.

## 3. Results

A total of 606 children admitted to pediatric intensive care units with severe acute respiratory infection (SARI) were screened for the viral panel. The result of the viral panel identified 446 children with one infected with a single virus and 160 co-infected with two or more viruses. Among the viruses detected, Human Rhinovirus (hRV) was the most prevalent (30%), followed by Respiratory Syncytial Virus RSV (17%), SARS-CoV-2 (13%), and coinfections 24% (Figure 1). 

Viral coinfections accounted for 24% of all samples studied, and among these, 79.4% of the patients analyzed had coinfections with only two viruses, and 20.6% of the affected patients had coinfections caused by three or more viruses (multiple virus). 

This study employed descriptive analyses where a total of twenty-two coinfections among SARI-causing viruses were identified. Thus, the five most frequent coinfections that were selected for the study are: hRV/SARS-CoV-2 (17.91%), hRV/RSV (14.18%), RSV/SARS-CoV-2 (12.69%), hRV/BoV (10.45%), and hRV/AdV (8.21%) (Figure 2).

It was analyzed which coinfections were more prevalent among age groups, and it was verified that the most affected of the public were children aged between 0.1 and 2 years old, with virus coinfections such as RSV/SARS-CoV-2 (7%) for 0.1 and 2 years old and hRV/RSV (9.1%) for 0.1 and 2 years old (Figure 3). It is worth mentioning that the coinfection caused by hRV/SARS-CoV-2 also stands out, where it was possible to verify its involvement in the age groups of 0–2 years (5.6%), 2–5 years (6.6 %), and 5–14 years old (3%) (Figure 3).

It was also evaluated whether there were differences in the length of stay of patients in intensive care units (ICU), in patients who were hospitalized with two viral types, and patients who were co-infected with several types of viruses. Patients co-infected with two viral types were hospitalized for a period of 161 h (6.7 days) and children co-infected with several viral types remained hospitalized for a period of 147 h (6.1 days), but despite verifying a reduction in the length of hospital stay of patients affected by multiple types of viruses, there were no statistically significant differences when compared with the group co-infected with only two types of viruses (*p* > 0.005) (Figure 4).

The age range of coinfected patients has been analyzed based on data obtained previously. Thus, the most significant age group was 38.1%, representing patients aged between 24 and 59 months (61 individuals). Patients older than 59 months represented a total of 27.5%, comprising forty-four patients (Figure 5A). Then, evaluations of the age groups affected by the five most frequent coinfections were carried out. For the group representing co-infections caused by hRV/SARS-CoV-2, the age group with the highest frequency (33.3%) was patients aged over 59 months (Figure 5B).

For the group of patients co-infected with RSV/SARS-CoV-2, three groups showed the same rates of frequency of involvement. Thus, patients younger than 6 months, between 24 to 59 months, and older than 59 months represented an individual percentage of 25% for each group (Figure 5C). The most representative age group of patients affected by hRV/RSV was from 24 to 59 months, totaling 68.75%. The other age groups had low frequencies of coinfection (Figure 5D). When analyzing the coinfections caused by hRV/hBoV, the most affected age group was those aged over 59 months, representing a total of 46.2% of the cases (Figure 5E). The highest frequency of patients affected by hRV/AdV were those aged between 24 to 49 months, which totaled 50% of cases (Figure 5F).

Another variable evaluated was the percentage of patients who required Invasive Mechanical Ventilation (IMV) and Non-Invasive Mechanical Ventilation (NIV): 16.25% of co-infected patients underwent IMV and 18.13% underwent NIV (Table 1).

When the duration of oxygen therapy was observed for patients with different coinfections, only patients with Rhinovirus/SARS-CoV-2 coinfection had a shorter oxygen therapy time when compared to patients with Rhinovirus/Bocavirus (*p* < 0.001), Rhinovirus/CoVs (*p* < 0.01), Rhinovirus/Metapneumovirus (*p* < 0.02), and Rhinovirus/VSR (*p* < 0.02). All patients with the other different coinfections had similar oxygen therapy use time (*p* > 0.05) (Figure 6).

Although the number of hospitalization days between those infected with RSV and non-infected with this virus did not have a statistically significant difference (*p* > 0.36), we compared the number of hospitalization days of children co-infected with RSV and other children co-infected with other viruses and demonstrated that children coinfected with RSV spent more days in the hospital (7.6 days) than uninfected children (6 days) (Table 2).

A descriptive analysis demonstrated the clinical outcome for all cases related to the five most frequent coinfections; unfortunately, due to the severity of the infection, some children died. Thus, we saw that two of the patients co-infected with hRV/SARS-CoV-2 died, representing 9.52% of all deaths in the study. In addition, both hRV/hBoV and hRV/RSV had death records for each case, representing 8.33% and 6.67% of all deaths, respectively (Table 3).

We compared age groups versus clinical course, and identified eight deaths of five children aged over 59 months (3.13%), one aged between 12 and 23 months (0.63%), and two aged between 24 and 59 months (1.25%). For the proportion of deaths versus comorbidities, we observed that 50% of the dead children had some type of comorbidity, and neurological disease was the most prevalent. The other 50% of children who died were due to the severity of SARS symptoms caused by viruses (Table 4). 

The analysis was conducted between the years 2020 and 2021. In the year 2020, hRV/BoV was more frequent in relation to other types of coinfections, representing a total of 35.1%. Then, hRV/SARS-CoV-2 represented a total of 24.3% in relation to the other coinfections analyzed during this period. RSV/SARS-CoV-2, hRV/RSV, and hRV/AdV had a total of 16.2%, 13.5%, and 10.8% of coinfections, respectively (Figure 7A).

Moreover, the year 2021 presented a divergent profile, with hRV/SARS-CoV-2 coinfection being the most frequent (30.8%), followed by hRV/RSV (28.2%). Additionally, 25.6% and 15.4% represented coinfections between RSV/SARS-CoV-2 and hRV/AdV, respectively. Interestingly, hRV/BoV presented the highest rates of coinfection during the year 2020. However, this type of coinfection was not recorded in the year 2021 (Figure 7B).

Coinfection frequency levels have also been evaluated in relation to the months of the year. For the year 2020, the coinfection that presented the highest rate of involvement in the months of April, September, and October was hRV/BoV. Rhinovirus/SARS-CoV-2 also had elevated levels of coinfection for the months of April and August 2020. In the month of April, an exponential increase in coinfection caused by RSV/SARS-CoV-2 was observed until the month of June, followed by a decline in cases until the month of August (Figure 7C).

When analyzing the coinfection rates for the months of 2021, two main types of coinfections have been highlighted. Initially, there was an increase in coinfections caused by hRV/SARS-CoV-2 between January and March, followed by a decline until June. Subsequently, increases in rates of hRV/SARS-CoV-2 coinfections were again observed in August, and an exponential increase in hRV/RSV coinfections between April and May, a decrease in cases until June, and stabilizing coinfection rates up to October 2021 (Figure 7D).

## 4. Discussion

Since the beginning of the COVID-19 pandemic, several measures have been taken by the government of Brazilian states to assist the population with SARI suspected of being contaminated by SARS-CoV-2. Our findings are important to demonstrate the circulation of respiratory viruses and their interactions with other viruses that caused severe acute respiratory infection in children during the period of health mitigation to contain COVID-19.

In this study, we detected 22 variations with two viruses (79.4%) or three viruses or more (20.6%), and the most prevalent viral coinfections were hRV/SARS-CoV-2, hRV/RSV, RSV/SARS-CoV-2, hRV/BoV, and hRV/AdV. Silva et al. [10] studied viral panels in children of ages between 18 days and 13 years old. The results indicated that rhinoviruses, adenoviruses, metapneumoviruses, and influenza were among the most important agents of ARI in pediatrics; equal results were found in this work. In a study prior to the COVID-19 pandemic, coinfection rates varied. Paulis et al. [11] found a rate of 31.2% of viral coinfection in children and A. Martínez-Roig et al. [12] found a rate of 61.81% (36.36% with two, 16.10% and 9.35% with more than two viruses). Already during the pandemic, some studies have demonstrated lower rates of coinfection. Elen Vink et al. [9] found a 2% coinfection rate, which claimed to be a rare event during the pandemic. Varella et al. [13] described an 8.9% rate of viral coinfection in children hospitalized in the pandemic. Nihan Şık and colleagues [14] found a rate of 86% coinfection with hRV and enterovirus being the most prevalent.

The highest proportion of coinfection in this study was between hRV and SARS-CoV-2, which we believe to be due to the high spread of the two viruses in the same period. Varela et al. [13], in a study like ours, found a rate of 7.1% of coinfection of hRV with SARS-CoV-2 and described that the viral occurrence was totally indistinct in epidemiological levels during the COVID-19 pandemic, since hRV was the most frequent virus co-circulating with SARS-CoV-2 in hospitalized children. Le Glass et al. [15] also described SARS-CoV-2 coinfection with hRV as more frequent, with patients presenting more respiratory distress and a greater chance of being admitted to ICUs.

When we analyzed only the year 2020, the most frequent coinfection was hRV and hBoV. Wang et al. [16] detected a high rate of patients co-infected with hBoV and other viruses, with hRV and RSV being the most frequent, and suggested that hBoV is often associated with coinfection, stating that the acute infection was caused by this virus inducing responses in the immune system in children with respiratory diseases. HBoV can be detected alone in respiratory samples from children with acquired pneumonia, and the high incidence of severe pneumonia found in HBoV patients demonstrates that this virus is a major contributor to severe respiratory disease [17].

Li et al. [18] suggested that seasonal respiratory viruses decreased significantly in 2020, except for hBoV, which in late autumn and winter 2020 was very prevalent, compared to the last 10 years which was 0.7% to 13.3%, and at the beginning of the third wave of COVID-19 in South Korea, where the scenario was of intense social distancing measures.

In a study by Kim et al. [19], it was described that in 2020, there was a decrease in the circulation of some enveloped viruses such as influenza, metapneumovirus, parainfluenza, and RSV, but that there was a significant increase in the circulation of hRV and hBoV, which are non-enveloped viruses; these findings agree with our study.

Coinfection of RSV/hRV viruses was more prevalent in children aged 0 to 2 years, and the length of hospital stay was longer for children affected with RSV, although there was no statistically significant difference. These two viruses are the main agents that cause respiratory tract infection in children for all years [20,21].

Hanchi et al. [22], in research conducted in Marrakech, studied double or multiple coinfections and found a percentage of 38.8% of positivity in the tested samples. The authors indicated that there was a significant difference when comparing the pandemic period and the period before the pandemic (*p* < 0.001). They found a predominance of SARI during COVID-19. It was equal to our results; the most detected virus was HRV, followed by RSV. The authors believe that the results found before and during the COVID-19 pandemic could be explained by the impact of implementing preventive measures.

In a study carried out by Calvo et al. [23], the coinfections between hRV and RSV were in children around one year of age, and the ages have different characteristics in relation to the unique infections of each virus; where RSV affected babies up to one year of age and hRV was detected in children almost two years of age; and whether in simple infection or coinfection, contamination with RSV implies a worse prognosis with increased duration of both hypoxia and hospitalization days.

Costa et al. [24] described that single hRV infections were less severe, as they had a lower proportion of hospital admissions than hRV-RSV coinfections, and hRV-only patients were older than RSV patients with simple infection and coinfections. They concluded that severe illness in hRV infections is caused by other risk factors concurrent with risk factors for RSV-induced hospitalization.

The coinfection between RSV and SARS-CoV-2 affected all age groups, with a slightly greater difference in the 2- to 5-year-old age group, and it was observed that the use of oxygen therapy devices was statistically significant. Alvares [25], analyzing the hospitalization of 32 children of 2 years of age with COVID-19, found a rate of 18.7% of coinfection with RSV and that these children remained hospitalized longer.

Children are affected with respiratory infections by both SARS-CoV-2 and RSV, and coinfection with these two viruses can generate a serious problem in the scenario of the COVID-19 pandemic, as both viruses represent significant challenges in terms of diagnosis and treatment in children. It is not clear how severe the pathogenicity of SARS-CoV-2 is for children, but coinfection of RSV with SARS-CoV-2 can be serious and affect clinical prognosis, making treatment difficult [26]. 

In the hospitalization of children with severe SARS-CoV-2 infection, 32% required some form of respiratory support. RSV can cause severe respiratory illness in children that require hospitalization and, in some cases, can lead to death [25,26]; therefore, coinfection with the two viruses can worsen the clinical condition of the child with SARI. Despite not having a statistically significant result, Lee et al. [27] reported that SARS-CoV-2–positive patients with coinfection had a higher prevalence of hospitalization, with a significantly longer hospital stay. 

In this study, 3.75% of children died during the period of hospitalization. Of these children, 50% had some type of comorbidity, with an exceptionally long hospital stay, and neurological damage was the highest rate.

In three patients who died, there was SARS-CoV-2 coinfection with other viruses, two of whom had underlying neurological diseases and one who had an endocrine disease. Some studies have shown that there is an extremely high association between a severe course of COVID-19 in children and some risk factors including cancer, immunodeficiency, chronic lung and heart disease, genetic and neurological diseases, diabetes, and obesity [28,29]. In a literary review conducted by Tsankov et al. [30], they examined the severity of COVID-19 infection among pediatric patients with comorbidities and concluded that children with pre-existing conditions are at increased risk of severe COVID-19 and associated mortality.

Most children do not require hospitalization for COVID-19 as SARS-CoV-2 causes mild symptoms in this age range, but the need for ICU admission depends on the comorbidities and complex medical history of the children evaluated [31,32]. The influence of comorbidities and SARS-CoV-2 infection on clinical outcome may be combined, and this virus may exacerbate coexisting chronic disease or be an additional factor in the severity of a clinical outcome in a patient [31].

In Brazil, a total of 1439 children under 5 years of age died because of COVID-19 during the first two years of the pandemic. For these deaths, SARS-CoV-2 made the infection worse for cases with pre-existing risk conditions and was associated with the leading cause of death; that is, SARS-CoV-2 aggravated a pre-existing problem [33].

A child who had no comorbidity had brain death decreed after 9 days of hospitalization, due to RSV/hRV coinfection, where he developed hypoxic encephalopathy and severe ischemia. Wouk et al. [34], in a literature review study, concluded that the appearance of neurological disorders in viral infections can cause neurological symptoms or lead to immune responses that trigger these pathological signs, since irreversible damage and cell death can occur if there is chronic dysfunction of the cells. Neuronal cells, both central and peripheral, can influence the development and progression of these disorders.

Neurological disorders can be caused by pathogens that cross the intact blood barrier, causing severe encephalitis or acute infections progressing to chronic disease or leading to death; these disorders can also occur indirectly through accumulation of protein aggregates, elevated levels of oxidative stress, alterations in autophagic mechanisms, synaptopathy, and neural destruction [33,35,36].

The seasonality of coinfections was evaluated, demonstrating that there was a difference between the years 2020 and 2021. In 2020, hRV/hBoV were detected in April, September, and October, and hRV/SARS-CoV-2 and RSV/SARS-CoV-2 detected in the months of April and August. In 2021, hRV/SARS-CoV-2 coinfection had peaks from January to March and August, and hRV/RSV peaked between April and May.

The epidemiological bulletin for COVID-19, in December 2020 in the state of Goiás/Brazil, showed the distribution of confirmed cases with severe acute respiratory infection (SARI) by date of hospitalization and moving average, where cases started in March and there were high peaks in June, July, and August [37]. In 2021, with the new wave of SARS-CoV-2, it did not differ from the previous year; the hospitalization rate increased in January, growing significantly in March [38].

Our study shows that more children in 2021 were infected with SARS-CoV-2, but it was also a period when mitigation measures were more lenient, and children were also returning to school.

We emphasize that according to research, in Brazil, there have been changes in the frequency of SARS-CoV-2 strains. At the beginning of the pandemic, strains B.1.1.28 and B.1.1.33 were more prevalent until October 2020, and then there was a predominance of two Brazilian variants, P.1 and P.2, originating from the lineage B.1.1.28. In just four months, these two Brazilian variants represented 75% of sequencing in Brazil. In terms of public health in Brazil, during the study period, four variants classified as variants of concern (VOC) and two (Zeta and Lambda) of the seven variants classified as variants of interest (VOI) by the WHO [39,40,41,42] were registered.

This study had limitations. The first limitation could indicate a bias in the results, the fact that we did not have viral panel data of the children evaluated in the outpatient clinic or ward. However, we believe that our data represent important data and do not agree with bias, since all were hospitalized with SARI and a viral panel was performed in all children. Because it was a pandemic season, it was recommended that children with mild flu symptoms stay at home. Another limitation would be to prove whether a co-infected child had a worse prognosis than children with single virus infection. Data such as blood tests and X-rays were not described in a cohesive manner in all medical records analyzed.

## 5. Conclusions

Coinfections with respiratory viruses, such as RSV and hBoV, can increase the severity of the disease in children with SARI who are admitted to the ICU, and children infected with SARS-CoV-2 have their clinical condition worsened when they have comorbidities.

## Figures and Tables

**Figure 1 biomedicines-11-01402-f001:**
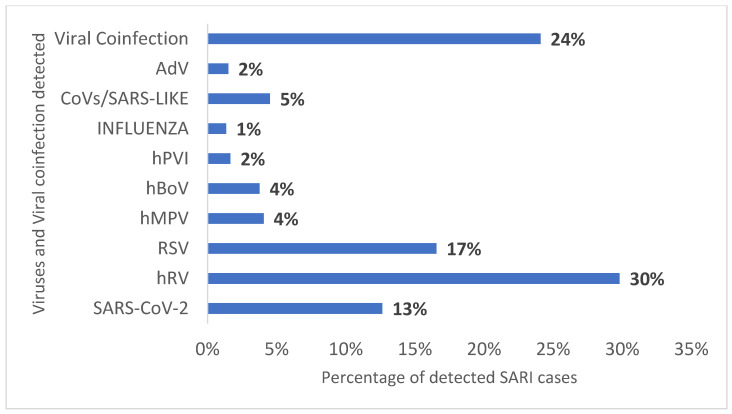
**Profile of the virus found in the viral panel of 606 children hospitalized with SARI in the years 2020 and 2021 in a tertiary hospital.** Legend: hRV (Human Rhinovirus), RSV (Respiratory Syncytial Virus), hMPV (Human Metapneumovirus), hBoV (Human Bocavirus), hPVI (Human Parainfluenza), CoVs (Coronavirus), and AdV (Adenovirus).

**Figure 2 biomedicines-11-01402-f002:**
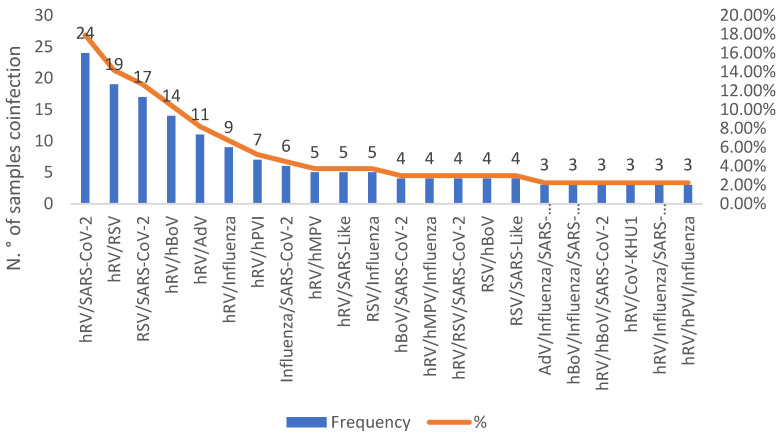
**Comparative analysis of twenty-two respiratory coinfections caused by viruses.** Legend: (hRV—Human Rhinovirus; HBoV—Human Bocavirus; AdV—Adenovirus; RSV—Respiratory Syncytial Virus; FLU—Influenza; hPVI—Human Parainfluenza Virus; CoV—Coronavirus; hMPV—Human Metapneumovirus).

**Figure 3 biomedicines-11-01402-f003:**
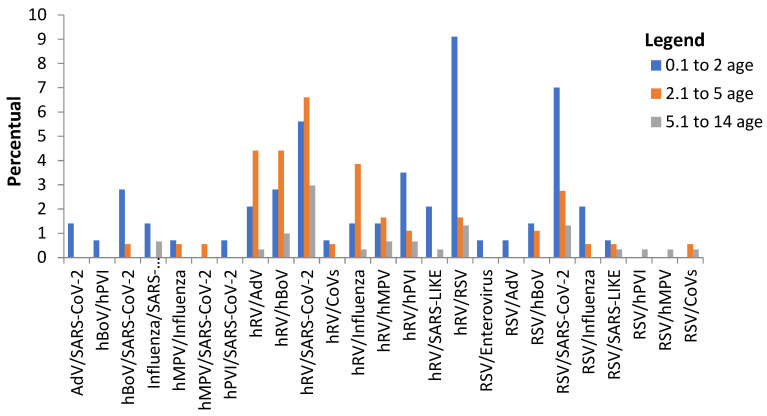
**Analysis of involvement of coinfections according to age group.** Legend: 0,1–2 years (blue), 2,1–5 years (orange), and 5,1–14 years (gray). Rhinovirus (hRV), Bocavirus (hBoV), SARS-CoV-2, Respiratory Syncytial Virus (RSV), Adenovirus (AdV), Coronavirus (CoVs), Human Metapneumovirus (hMPV), Parainfluenza (hPVI).

**Figure 4 biomedicines-11-01402-f004:**
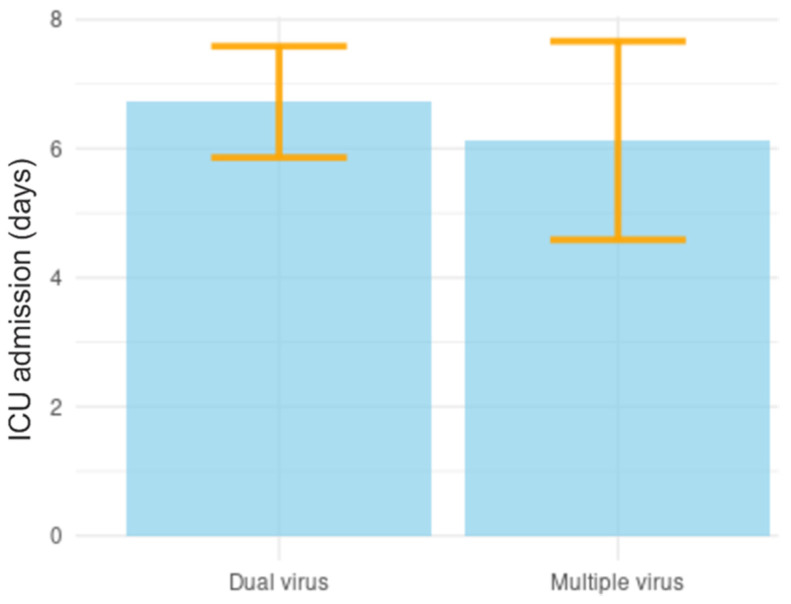
**Analysis of length of stay in intensive care units (ICU) caused by viral coinfections**.

**Figure 5 biomedicines-11-01402-f005:**
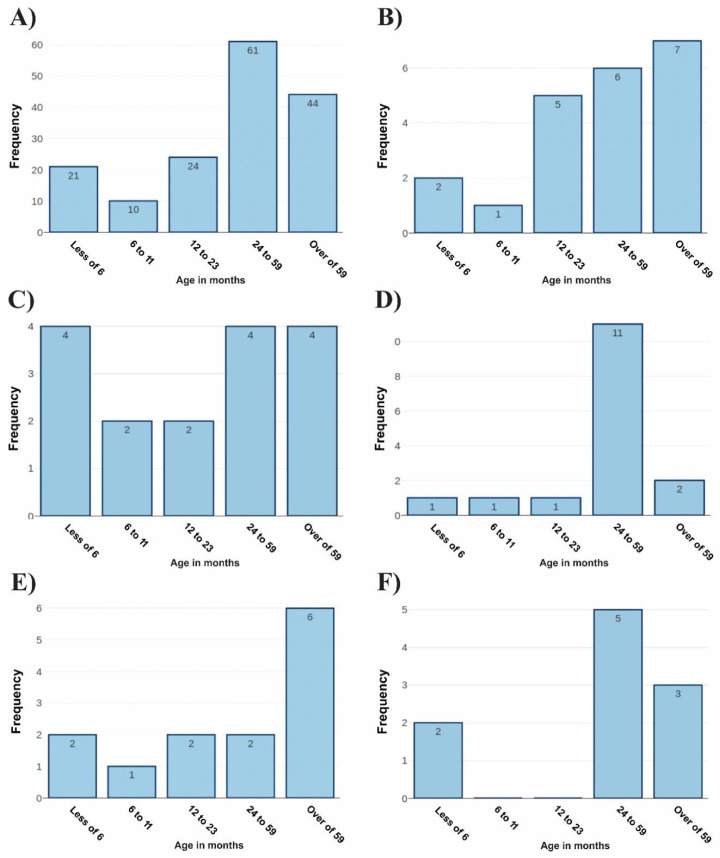
**Analysis of the age group of co-infected patients. Legend:** (**A**) representation of the age group of patients in relation to all coinfections analyzed during the study. (**B**) The age range of patients co-infected with hRV and SARS-CoV-2. (**C**) The age group of individuals co-infected with RSV and SARS-CoV-2. (**D**) The age group of patients co-infected with Rhinovirus and RSV. (**E**) Individuals affected by hRV and hBoV. In (**F**), individuals affected by coinfections caused by hRV and AdV had their age profile highlighted.

**Figure 6 biomedicines-11-01402-f006:**
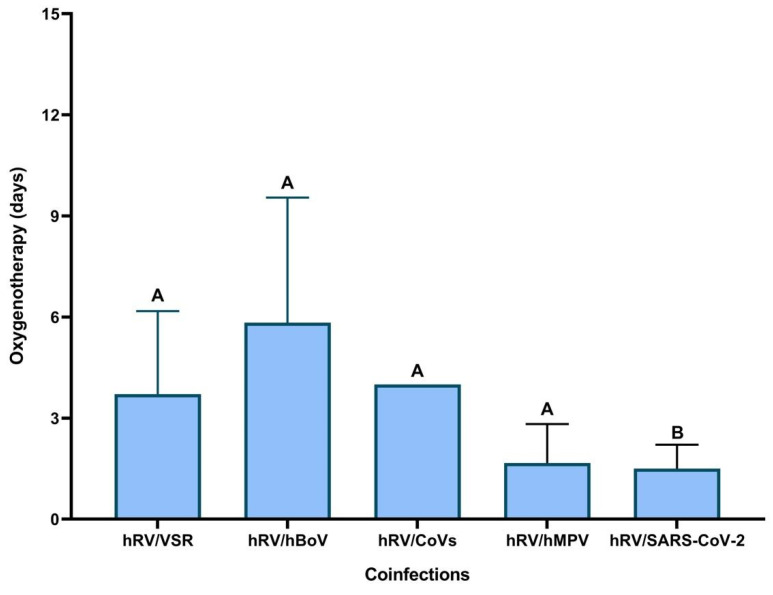
Time of use of oxygen therapy in children with viral coinfection with SARI. (hRV—Rhinovirus; hBoV: Bocavirus; CoVs: Coronavirus; hMPV: Metapneumovirus). Legend: letter (**A**) means all the virus types were statistically significant with relation to oxygenotherapy; letter (**B**) the virus type have no significant statistic with relation to oxygenotherapy.

**Figure 7 biomedicines-11-01402-f007:**
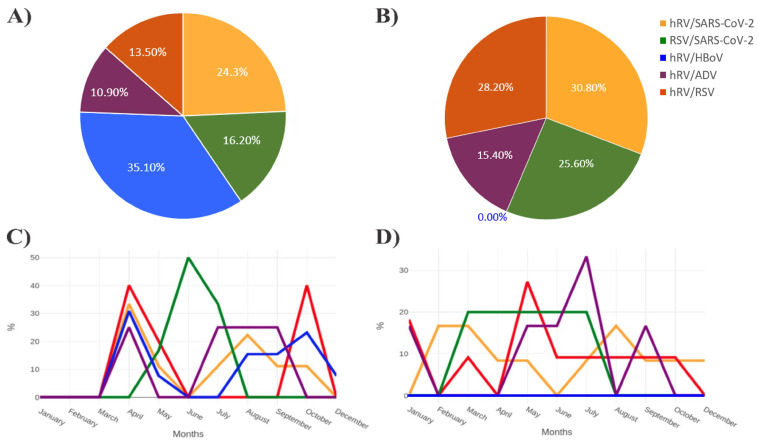
**Analysis of the seasonality profile of the most frequent viral coinfections in the years 2020 and 2021.** Legend: (**A**) the percentage of the five most frequent coinfections in 2020 is observed. (**B**) Refers to the percentages of the five most frequent coinfections in 2021. (**C**) Analysis of the frequency of coinfections in relation to the months of the year in 2020. (**D**) Characterization of the most frequent coinfections in relation to the months of 2021.

**Table 1 biomedicines-11-01402-t001:** Analysis of the need for IMV and NIV during ICU stay.

Variables	IMV	NIV
**YES**	26 (16.25%)	29 (18.13%)
**NO**	126 (78.75%)	123 (76.88%)
**N/A**	8 (5%)	8 (5%)

Invasive Mechanical Ventilation (IMV); Non-Invasive Mechanical Ventilation (NIV).

**Table 2 biomedicines-11-01402-t002:** Comparison between the length of stay of patients co-infected with RSV in the ICU and those not co-infected with RSV.

Statistic/Coinfection	Coinfection with RSV	Coinfection without RSV
**Average ICU days**	7.69	6.00
**Standard Error**	0.72	1.65
***p*-value**	0.36	

**Table 3 biomedicines-11-01402-t003:** Descriptive analysis of the clinical outcome of co-infected patients.

Outcome	General	hRV/ SARS-CoV-2	RSV/ SARS-CoV-2	hRV/ RSV	hRV/ hBoV	hRV/ AdV
**Hospital discharge**	152 (95.0%)	19 (90.48%)	11 (100%)	14 (93.34%)	11 (91.67%)	10 (100%)
**Death**	8 (5.00%)	2 (9.52%)	0 (0.0%)	1 (6.67%)	1 (8.33%)	0 (0.0%)

Legends: hRV—Human Rhinovirus; RSV—Respiratory Syncytial Virus; hBoV—Human Bocavirus; AdV—Adenovirus.

**Table 4 biomedicines-11-01402-t004:** Characteristics of patients with coinfection by two or more viruses who died during the COVID-19 pandemic.

Age/Months	Comorbidities	*ICU Days	*IMV–O2 Days	Viruses
20	Neurological	22	19	SARS-CoV-2/Influenza/ Rhinovirus
31	Neurological	15	15	SARS-CoV-2/Rhinovirus
36	No comorbidity	9	9	RSV/Rhinovirus
63	Neurological	20	18	Bocavirus/Rhinovirus
84	No comorbidity	9	9	RSV/Metapneumovirus
108	No comorbidity	26	25	Rhinovirus/Metapneumovirus
132	No comorbidity	1	1	SARS-CoV-2/RSV
144	Endocrine	7	7	SARS-CoV-2/Influenza/ Rhinovirus

Legend: IMV—Invasive Mechanical Ventilation; O2: oxygen; ICU—Intensive Care Unit.

## Data Availability

Data sharing is not applicable to this article as no new data were created or analyzed in this study.

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
