# Peer review of "Viral Coinfection of Children Hospitalized with Severe Acute Respiratory Infections during COVID-19 Pandemic"

_biomedicines, 2023, doi:10.3390/biomedicines11051402_

Round 1

Reviewer 1 Report

Ito et al., described the situation of dual and multiple respiratory virus infections in pediatric inpatient in ICU in the Center-West region of Brazil from 2020 to 2021. The importance of the study can be recognized, but the manuscript seems to be difficult to understand.

Comments:

1.      Overall, the usage of “SARS” was very confusing. In L130, “SARS” was defined as the abbreviation of "severe acute respiratory infection” but in L294-295 “severe acute respiratory syndrome” was also used. What the “SARS” mean? Now “SARS” can be read as “SARS-CoV-2 infection”, therefore, there are some difficult sentences to be understood. I wonder why “SARS” was used instead of “SARI”. The definition of each technical term should de re-checked and the entire manuscript should be re-written carefully.

2.      The data of single infections should be added as the reference for the dual/multiplex infections. The author group already published the data (Eur J Clin Microbiol Infect Dis 2022. 41:1445-1449). The study must be cited appropriately, and the comparison should be performed to enhance the discussion.

3.      Table 1 and 2 are very difficult to understand. The data of ventilation is important; therefore, the tables should be shown easy to be understood.

4.      The discussion part seems diffused and there are unnecessary citations. Discussing the co-infection with hRV is important, but, there are other points should be discussed. The specimens collected from March 2020 to December 2021. This means strains of SARS-COV-2 are largely different in first half (Wuhan type) and second half (various VOCs). The difference depending on SARS-CoV-2 strains can be discussed. In 2020, RSV, Flu, hMPV infections were extremely lower in northern hemisphere. The difference of situation of these infections in Brazil from other countries can be discussed.

Minor comments:

L35: “Conclusion:” remains.

L100: The kit name should be stated.

L103-117: The amplification protocol seems not to be fixed. Are these different for each target? Is the cut-off value following the instructions? If the authors modified the protocols of IVD (or RUO) kit by themselves, the validation of the protocol is required. The data should be stated or cited correctly. If the kit name is known, this part can be compressed as “following the manufacture’s instructions”.  

Figure 6: “Over of 29”?

L212: “p-value=013”?

Figure7: “hRV/ADE”?

Author Response

Reviwer 1

Reviewer: Overall, the usage of “SARS” was very confusing. In L130, “SARS” was defined as the abbreviation of "severe acute respiratory infection” but in L294-295 “severe acute respiratory syndrome” was also used. What the “SARS” mean? Now “SARS” can be read as “SARS-CoV-2 infection”, therefore, there are some difficult sentences to be understood. I wonder why “SARS” was used instead of “SARI”. The definition of each technical term should de re-checked and the entire manuscript should be re-written carefully.

Authors answer: The authors changed the SARS for SARI in all text.

Reviewer: The data of single infections should be added as the reference for the dual/multiplex infections. The author group already published the data (Eur J Clin Microbiol Infect Dis 2022. 41:1445-1449). The study must be cited appropriately, and the comparison should be performed to enhance the discussion.

Authors answer: The author added the citation and marked of yellow.

Reviewer:  Table 1 and 2 are very difficult to understand. The data of ventilation is important; therefore, the tables should be shown easy to be understood.

Authors answer: The tables were removed.

Reviewer: The discussion part seems diffused and there are unnecessary citations. Discussing the co-infection with hRV is important, but, there are other points should be discussed. The specimens collected from March 2020 to December 2021. This means strains of SARS-COV-2 are largely different in first half (Wuhan type) and second half (various VOCs). The difference depending on SARS-CoV-2 strains can be discussed. In 2020, RSV, Flu, hMPV infections were extremely lower in northern hemisphere. The difference of situation of these infections in Brazil from other countries can be discussed.

Authors answer: An article was added, the difference situation were discussed and are in discussion, marked of yellow.

Minor comments:

Reviewer:  L35: “Conclusion:” remains.

Authors answer: ok

Reviewer: L100: The kit name should be stated.

Authors answer: The kit name were added.

Reviewer: L103-117: The amplification protocol seems not to be fixed. Are these different for each target? Is the cut-off value following the instructions? If the authors modified the protocols of IVD (or RUO) kit by themselves, the validation of the protocol is required. The data should be stated or cited correctly. If the kit name is known, this part can be compressed as “following the manufacture’s instructions”. 

Authors answer: Are different target, but the amplification protocol is fixed by Thermo Fisher®. The protocol details were changed by the phrase: “following the manufacture’s instructions”.

Reviewer: Figure 6: “Over of 29”?

Authors answer: “Over 59” years means patients over 59 years of age; "age in months", as written on the x-axis of the figure 5.

Reviewer: L212: “p-value=013”?

Authors answer: The informations was stardarded.

Reviewer: Figure7: “hRV/ADE”?

Authors answer: The writen was corrected.

Reviewer 2 Report

1. Please check and edit English language and style throughout the manuscript

2. The title should be revised ("RESPIRATORY" instead of "PESPIRATORY"). And I suggest correcting it as "Severe Acute Respiratory Syndrome"

3. Line 85: Please define your studied population (suspected respiratory virus infection) for more clear

4. The title of the article does not match the research object, because your work was conducted on sick children with suspected viral respiratory infections. But the title is about severe acute respiratory syndrome

5. The target pathogens are in a multiplex PCR? If not, please explain your choice of these viruses

6. I suggest adding a supplementary data on the results of co-infection for more details

7. Please add the numerator and denominator with each proportion in Results section. 

For example, in the Figure 2, we don't know the proportion of co-infections was calculated on which number of patients

8. All patients were positive for at least one pathogen?

Author Response

Revisor 2

Reviewer:  Please check and edit English language and style throughout the manuscript

Authors answer: The English language was reviewed.

Reviewer: The title should be revised ("RESPIRATORY" instead of "PESPIRATORY"). And I suggest correcting it as "Severe Acute Respiratory Syndrome"

Authors answer: The title was changed.

Reviewer: Line 85: Please define your studied population (suspected respiratory virus infection) for more clear

Authors answer: The phrase was corrected.

Reviewer: The title of the article does not match the research object, because your work was conducted on sick children with suspected viral respiratory infections. But the title is about severe acute respiratory syndrome

Authors answer: The title was changed.

Reviewer: The target pathogens are in a multiplex PCR? If not, please explain your choice of these viruses

Authors answer: The pathogen were monoplex PCR. This viral painel is routinely done on Brazil health public; this is the reason of the our choice.

Reviewer: I suggest adding a supplementary data on the results of co-infection for more details

Authors answer: One of the reviewer asked for take of the supplementary material.

Reviewer:  Please add the numerator and denominator with each proportion in Results section. For example, in the Figure 2, we don't know the proportion of co-infections was calculated on which number of patients. All patients were positive for at least

Authors answer: The information is below figure: “Viral co-infections accounted for 24% of all samples studied, and among these 79.4% of the patients analyzed, had coinfections with only two viruses, and 20.6% of the affected patients had co-infections caused by three or more viruses (multiple virus)”.

Round 2

Reviewer 1 Report

The revised manuscript still contains some concerns.

In Table 1, comparison with the data in single infection is required. It can show the relation between diseases severity and multiple infections, which is the purpose of this study written in last sentence in Introduction.

The chart legend in figure 7 remains incorrect.

The discussion section is too long. The descriptions of unnecessary citations can be removed.

Author Response

Reviwer 1

  1. English language and style are fine/minor spell check required

Authors answer: The English spell check was marked up using the “Track Changes”.

  1. In Table 1, comparison with the data in single infection is required. It can show the relation between diseases severity and multiple infections, which is the purpose of this study written in last sentence in Introduction.

Authors answer: The authors dont have this statistical results. To add the comparison with the data in single infection, is necessary do the statistical comparation between each other virus indivivualy.

3.The chart legend in figure 7 remains incorrect.

Authors answer: The correction was done.

  1. The discussion section is too long. The descriptions of unnecessary citations can be removed.

Authors answer: The authors believe that all the citations are important because the article has a lot of results and is important discuss all them. We hope that the reviewer should apoint the specific references that he think is not important and the authors will delete it according with the suggestions.

Reviewer 2 Report

Thank you for your responses

Author Response

English language and style are fine/minor spell check required

Authors answer: The English spell check was marked up using the “Track

Changes”.

Round 3

Reviewer 1 Report

This study used almost same group of patient in study reported by your group (Eur J Clin Microbiol Infect Dis 2022. 41:1445-1449). The description in ”Target population” is copy paste from the manuscript. I wonder why the author stubbornly don’t discuss with the single infections in previous study. The Silva’s study should be cited in method section, and it is better to discuss the diseases severity of multiple infection relative to single infection based on not the citation but the same patient group, citing Silva’s study.

The author should check the image of figure 7 carefully, especially the legend in the right of B). I think “hRV/ADE” was never modified from the original submission. Otherwise, does it mean adenovirus E?